# OpenReview forum: "TRUST-VLM: Thorough Red-Teaming for Uncovering Safety Threats in Vision-Language Models"
_ICML.cc/2025/Conference — ICML 2025 poster_

### Official Review · Reviewer_YghG · 2025-03-07

**Overall Recommendation:** 3

**Summary:**

This paper presents a framework named TRUST-VLM for automatic red-teaming vision-language models. The framework mainly involves three stages of test-case generation, execution and evaluation, and test-case refinement, by incorporating a large language model and a text-to-image model. It is shown to be more effective than static datasets.

## Update After Rebuttal
After reading the responses to all the reviewers, most of my concerns have been addressed. Therefore, I will raise my rating accordingly. I highly suggest the authors include the discussions above in their revision to highlight their technical novelty and insights.

**Claims And Evidence:**

Yes. The work is well-motivated with clear statements. However, practical recommendations for improving VLMs mentioned in the contributions are not sufficiently discussed in the main text.

**Essential References Not Discussed:**

Besides HADES, there are also some other works associated with red-teaming or adversarial attacks on VLMs that should be reviewed and discussed. Some are listed below.

Dong et al., How Robust is Google's Bard to Adversarial Image Attacks?
Gong et al., Figstep: Jailbreaking large vision-language models via typographic visual prompts.
Zhang et al., MultiTrust: A Comprehensive Benchmark Towards Trustworthy Multimodal Large Language Models

**Experimental Designs Or Analyses:**

The authors mainly compare TRUST-VLM with two static datasets, JailbreakV-28K and RTVLM. There are also other automatic red-teaming methods against VLMs, like HADES (ECCV2024), which should be compared with.

Also, the target activities in different datasets or methods are not restricted in the same domain. A discussion about the statistics or characteristics of auto-generated text prompts would help guarantee a fair comparison and further improve the analysis.

**Methods And Evaluation Criteria:**

The pipeline contains necessary modules for iterative red-teaming. The evaluation considers not only the effectiveness of test cases, but also the diversity, alignment and toxicity, which are comprehensive.

**Other Comments Or Suggestions:**

See above.

**Other Strengths And Weaknesses:**

Strength:
* The writing is clear to follow.
* Automatic red-teaming is a significant issue to study.
* The results of TRUST-VLM are effective as shown by extensive experiments.

Weakness:
* The proposed framework is only compared with two static datasets, not showing the advantage of auto red-teaming, which is not convincing enough.
* The framework highly relies on the overall design by human, like prompts, pipelines, categories, rather than proposing novel algorithms. It also provides few practical insights into how to better reinforce VLMs.

**Questions For Authors:**

See above.

**Relation To Broader Scientific Literature:**

The paper proposes an automatic framework for red-teaming VLMs, which can even effectively stress-test closed-source models. However, the work is mainly based on prompt engineering and artificial design. There is limited insight in either red-teaming techniques or certain flaws in modern VLMs.

**Theoretical Claims:**

N/A

---

> ### Author Rebuttal · Authors · 2025-04-01
>
> Thanks for your constructive feedback and suggestions.
>
> >**Q1: Comparison with automatic red-teaming methods like HADES.**
>
> R1: Thank you for your thoughtful suggestion. We have conducted comparisons with both HADES and another recent jailbreak-based method. Besides, we also conducted comparisons with the SOTA red-teaming method, Arondight. Due to space constraints, we kindly refer you to our detailed responses to Reviewer iB5W’s Q2 and Q3, where we analyze these methods and contrast their performance with our TRUST-VLM approach.
>
> >**Q2: Reliance on human-designed prompts and categories.**
>
> R2: In this paper, our primary objective is to investigate whether we can effectively leverage the inherent capabilities of LLMs for red-teaming without relying on extensive data collection or computational resources. To this end, we employ demonstrations and iterative feedback mechanisms, enabling the LLM to autonomously generate diverse and effective adversarial test cases. This approach offers flexibility across models of varying modalities and architectures. Furthermore, recognizing that model developers might prioritize different safety aspects, our framework inherently allows easy adjustment and targeted exploration of various safety domains. Our experimental results also demonstrate that TRUST-VLM achieves strong performance across multiple models, diverse harmful categories, and varying generation settings, underscoring the general applicability and robustness of our auto red-teaming approach. We will explicitly clarify these advantages and generalizability of our auto red-teaming framework in our revision.
>
> >**Q3: Practical recommendations for reinforcing VLMs insufficiently discussed.**
>
> R3: We agree that discussion of defensive strategies enhances the practical impact of our research. Specifically, we provide two possible practical defensive approaches based on the adversarial test cases (both image and text prompts) discovered by our TRUST-VLM method:
> 1. **Safety Alignment via Fine-tuning**:
> We can utilize the harmful test cases identified by our red-teaming approach to fine-tune VLMs, explicitly aligning harmful inputs with safe or prohibited responses. This method mirrors standard industry practices for model safety alignment, serving as a “safety patch” that effectively mitigates discovered vulnerabilities.
> 2. **Neuron Masking for Model Purification**:
> Inspired by the method proposed in [1], another viable defensive strategy involves examining neuron activations triggered by harmful prompts. Using adversarial inputs (image and text prompts) identified by TRUST-VLM, model owners can pinpoint neurons responsible for undesirable behaviors and subsequently mask or remove these neurons to purify and strengthen the robustness of the model.
> From the above defensive strategies, it is evident that employing advanced red-teaming methods to uncover diverse vulnerabilities is the key to enhance the effectiveness and robustness of model protection. We will integrate these possible defensive approaches into our revision.
>
> >**Q4:Essential References Not Discussed**
>
> R4:Thank you for your valuable suggestion. Our paper primarily focuses on red-teaming as a systematic method for vulnerability discovery, rather than on jailbreak-style adversarial attacks. As such, we selectively discussed a few jailbreak-related works in the current version. We acknowledge that additional relevant works—such as those by Dong et al., Gong et al., and Zhang et al.—can further enrich the related work section. We will include these works in our revision to provide a more comprehensive overview of existing adversarial and red-teaming approaches for VLMs.
>
> ---
> [1] Huang et al., Antidote: Post-fine-tuning Safety Alignment for Large Language Models against Harmful Fine-tuning.

---

> > ### Comment · Reviewer_YghG · 2025-04-02
> >
> > Thanks for your response. After reading the responses to all the reviewers, most of my concerns have been addressed. Therefore, I will raise my rating accordingly. I highly suggest the authors include the discussions above in their revision to highlight their technical novelty and insights.

---

> > > ### Author Response · Authors · 2025-04-03
> > >
> > > Thank you for your valuable feedback and for raising your score after our rebuttal. We appreciate your thoughtful suggestions and will update the paper accordingly to reflect the important discussions raised.

---

### Official Review · Reviewer_75F2 · 2025-03-13

**Overall Recommendation:** 4

**Summary:**

The paper presents an automate, iterative mechanism to red team vision language multimodal models. The approach consists of three parts: (1) text case generation, (2) attacking the VLM, and classifying the responses, and (3) refining the test cases using the mdoeration feedback. The first and the final step involve using an LLM (LLama3.1-instruct in this paper) with in-context learning to refine the attack prompts. The attack images are generated using captions generated by the LLM. The paper present results of the red teaming approach on several categories, and show high FDR as compared to non-iterative benchmark-based methods.

## update after rebuttal

The rebuttal addressed most of my concerns. Though the evaluation is still fairly limited, I believe the authors present a convincing best-effort attempt. I am therefore keeping my score.

**Claims And Evidence:**

The claims are well-supported with a fairly thorough analysis of the results.

**Essential References Not Discussed:**

The paper provides a thorough description of VLM red-teaming methods. It also addresses approaches like adversarial attacks.

**Experimental Designs Or Analyses:**

The experimental design is generally correct. However, it might be useful to conduct some trials with varying temperature of the target VLMs as an additional ablation study to analyse the effectiveness of the approach. In addition, as suggested above, restrcitng the number of ICL examples would also be a good test of the effectiveness of the refinement process.

**Methods And Evaluation Criteria:**

The paper presents comparisons with static datasets while itself being an iterative method. An interesting additional ablation would be to restrict the number of ICL examples to 1, and see if the attack is still successful. Otherwise, I do find the given experiments to be comprehensive.

**Other Comments Or Suggestions:**

None.

**Other Strengths And Weaknesses:**

Overall, I find the paper to be quite original in its approach towards using ICL and text to image generation towards breaking VLMs. The ablation experiments also provide interesting insights into the workings of jailbreaking methods.

**Questions For Authors:**

1. Have the authors considered finetuning the language model to specifically output red-teaming captions instead of using ICL?
2. Could the authors also discuss the effect of number of tips / context length in terms of ICL examples?  Perhaps one experiment could on restricting the number of ICL examples or tips.
3. Can the authors also discuss some mitigation methods for such redteaming attacks?

**Relation To Broader Scientific Literature:**

The paper is well-motivated and very relevant given the large scale deployment of VLMs. In terms of contribution, it improves upon existing static benchmarking methods by leveraging off-the-shelf LLMs and image generators to test VLMs. It also improves upon several axes including diversity, low-toxicity (and therefore low detectiblity by filters), and provides a metric (number of refinements) as a way to track model improvement. While Arondight (2024) also leverages similar tools with RL, the proposed appraoch only uses prompt injection and ICL, which appears to be enough.

**Theoretical Claims:**

Not applicable

---

> ### Author Rebuttal · Authors · 2025-04-01
>
> We appreciate your detailed feedback and supportive comments.
>
> >**Q1: Restricting the number of ICL examples / Tips would be a good test of the refinement process.**
>
> R1: Thank you for the insightful suggestion. We conducted additional experiments as per your recommendation. As shown in the table below, when the number of ICL examples is reduced from 8 (our default) to 1, the Fault Detection Rate slightly increases. However, this comes at the cost of reduced diversity, weaker semantic alignment, and higher toxicity in the generated test cases.
>
> This trade-off is expected: using a larger number of ICL examples guides the red-teaming LLM to explore a broader and more diverse set of vulnerabilities. In contrast, with only one ICL example, the model tends to focus on more direct adversarial behaviors (e.g., jailbreak-style attacks), leading to higher attack success but lower diversity and increased toxicity.
>
> We will include these results and analysis in our revision to highlight the importance of the number of ICL demonstrations in balancing red-teaming effectiveness and safety.
>
> | # of ICL examples | FDR  | Textual Diversity | Visual Diversity | Textual Alignment | Visual Alignment | Textual Toxicity | Visual Toxicity |
> |-------------------|------|-------------------|------------------|-------------------|------------------|------------------|-----------------|
> | 1                 | 97%  | 0.71              | 0.39             | 0.75              | 0.26             | 19%              | 60%             |
> | 8 (default)       | 95%  | 0.88              | 0.50             | 0.76              | 0.26             | 11.67%           | 51%             |
>
> Moreover, we conducted an ablation study by reducing the number of tips from the default setting of 2 to 1. The table below shows that with only one tip, the FDR remains high (96%)—even slightly higher than the default setting. However, similar to the findings with reduced ICL examples, this comes at the cost of reduced diversity and increased toxicity. Thus, more tips helps guide the LLM toward generating more diverse and safer adversarial test cases, rather than concentrating on a narrow class of high-toxicity attacks. It reinforces the importance of multi-faceted guidance (via tips and ICL examples) in supporting controllable and diverse red-teaming generation.
>
> | # of Tips     | FDR  | Textual Diversity | Visual Diversity | Textual Alignment | Visual Alignment | Textual Toxicity | Visual Toxicity |
> |---------------|------|-------------------|------------------|-------------------|------------------|------------------|-----------------|
> | 1             | 96%  | 0.73              | 0.36             | 0.73              | 0.27             | 22%              | 59%             |
> | 2 (default)   | 95%  | 0.88              | 0.50             | 0.76              | 0.26             | 11.67%           | 51%             |
>
> >**Q2: Varying temperature settings in target VLMs.**
>
> R2: Thank you for the suggestion. In our default setting, we set do_sample=False for reproducibility (noting a typo in the appendix where it was incorrectly written as true). Following your advice, we varied the VLM temperature to 0.5 and 1.5. As shown below, our method achieved even higher FDRs, confirming its robustness. This aligns with findings from [1], which report that higher temperature increases vulnerability. We will add this result and correct the appendix typo in the revised version.
>
> | VLM Temp       | do_sample=False | 0.5  | 1.5  |
> |----------------|-------------|------|------|
> | FDR            | 95%         | 100% | 100% |
>
> >**Q3: Finetuning LLM for red-teaming specifically.**
>
> R3: We appreciate this insightful suggestion. Indeed, fine-tuning language models specifically for red-teaming can be an effective approach, as demonstrated in prior works such as [2]. In our paper, the primary focus is exploring whether demonstrations and iterative feedback can effectively guide an LLM to autonomously generate high-quality adversarial test cases without the need for extensive training data collection or substantial computational resources. Nonetheless, fine-tuning-based methods are complementary to our approach. A promising future direction would be to first fine-tune a red-teaming model using collected data, and subsequently apply our demonstration- and feedback-based methodology to further enhance red-teaming performance. We will clarify this complementary relationship and highlight potential integration in our future work discussions. Thanks again for this insightful suggestion.
>
> >**Q4: Mitigation methods for red-teaming attacks.**
>
> R4: We agree defensive strategies enhance the practical impact of our research. Due to space constraints, we kindly refer you to our detailed responses to Reviewer YghG’s Q3.
>
> ---
> [1] Huang et al., Catastrophic Jailbreak of Open-source LLMs via Exploiting Generation.
>
> [2] Li et al., ART: Automatic Red-teaming for Text-to-Image Models to Protect Benign Users.

---

> > ### Comment · Reviewer_75F2 · 2025-04-02
> >
> > I thank the authors for their response. The rebuttal has clarified my questions, and I will keep my recommendation as an 'accept'.

---

> > > ### Author Response · Authors · 2025-04-03
> > >
> > > Thank you for your kind and supportive review. We are glad our rebuttal addressed your concerns, and we will incorporate the main points from this exchange into our revised paper.

---

### Official Review · Reviewer_Bw4z · 2025-03-14

**Overall Recommendation:** 4

**Summary:**

This paper introduces TRUST-VLM, a novel multi-modal automatic red-teaming approach that leverages in-context learning and target model feedback to enhance attack success rates and test case diversity. Experimental results show that TRUST-VLM surpasses traditional methods, offering actionable insights for improving VLM safety.

## Update after rebuttal
I raised my recommendation to accept as the rebuttal addressed my concerns.

**Claims And Evidence:**

Yes.

**Essential References Not Discussed:**

Please refer to the references listed above.

**Experimental Designs Or Analyses:**

**Experiments**:

1. **Insufficient test cases.** The evaluation with 200 test cases is not thorough or reliable enough. Authors are encouraged to conduct more extensive attacks, ideally scaling up to thousands of test cases.

2. **Limited comparison with baselines.** The proposed method is only compared against baselines using a single model, LLaVA, in Table 1. To demonstrate generalizability, similar comparisons should be conducted on more diverse model architectures.

3. **Missing comparison with state-of-the-art methods.** The study lacks a comparison against the latest multimodal jailbreaking approaches [3, 4]. Including these baselines would help position the contribution relative to existing work.

4. **Lack of evaluation details.** It is unclear how many runs were conducted for the reported results. If multiple runs were performed, the authors should report standard deviations to assess the stability and variance of their method.

**Reference**:

[3] Shayegani, Erfan, Yue Dong, and Nael Abu-Ghazaleh. "Jailbreak in pieces: Compositional Adversarial Attacks on Multi-Modal Language Models." The Twelfth International Conference on Learning Representations. 2023.

[4] Qi, Xiangyu, et al. "Visual adversarial examples jailbreak aligned large language models." Proceedings of the AAAI conference on artificial intelligence. Vol. 38. No. 19. 2024.

**Methods And Evaluation Criteria:**

**Methods**:

1. **Novelty is limited.** The idea of using LLMs to automate red-teaming has been explored in previous works. However, the proposed method appears to be sound. The implementation of the feedback mechanism through entailment-based classification is particularly interesting.

2. **Contribution of the input image to jailbreaking is unclear.** Based on the samples provided in the appendix (pages 20-23), the success of jailbreaking does not seem strongly dependent on the input image. For example, in Figure 20, the VLM output explicitly states, *"I'm unable to see or analyze images."* To clarify this, authors are encouraged to test the sensitivity of the method to the input image—for instance, by replacing the current image with a random image or an alternative image describing the same activity.

**Reference**:

[1] Deng, Boyi, et al. "Attack Prompt Generation for Red Teaming and Defending Large Language Models." The 2023 Conference on Empirical Methods in Natural Language Processing.

[2] Li, Guanlin, et al. "ART: Automatic Red-teaming for Text-to-Image Models to Protect Benign Users." The Thirty-eighth Annual Conference on Neural Information Processing Systems.

**Other Comments Or Suggestions:**

No.

**Other Strengths And Weaknesses:**

No.

**Questions For Authors:**

No.

**Relation To Broader Scientific Literature:**

It presents a new method to automate red-teaming of VLMs, achieving a higher success rate while being less detectable.

**Theoretical Claims:**

No theoretical claims.

---

> ### Author Rebuttal · Authors · 2025-04-01
>
> Thank you very much for your valuable feedback and insightful suggestions.
>
> >**Q1: Limited novelty of LLM-based automation.**
>
> R1: We acknowledge that various red-teaming methods have emerged recently, targeting different models, such as LLMs [1] and text-to-image models [2]. However, these existing methods cannot be directly applied to Vision-Language Models (VLMs), as VLMs inherently operate with two modalities (image and text) that exhibit complex inter-modal correlations (as further illustrated in our response to Q2 below). To the best of our knowledge, the only existing red-teaming method specifically designed for VLMs is Arondight. However, Arondight requires reinforcement learning for training, whereas our TRUST-VLM leverages in-context learning, offering a significantly more lightweight yet effective solution. Experimental results demonstrate that TRUST-VLM achieves superior performance in terms of both attack success rate and diversity of discovered vulnerabilities. Due to space constraints, we kindly refer the reviewer to our response to Reviewer iB5W’s Q3 for details of this comparison.
>
> We will clearly emphasize these distinctions and explicitly highlight the advantages of our lightweight ICL-based feedback mechanism in comparison to existing RL-based methods in our revision.
>
> ---
> >**Q2: Unclear contribution of images to jailbreaking success.**
>
> R2:  Thank you for highlighting this important point. Following your suggestion, we conducted additional experiments to assess the sensitivity of our method to the input image. Specifically, we combined an optimized textual prompt with two types of images: (1) a random image, and (2) an image from a different optimization round but within the same harmful activity category.
> The results are as follows:
> - Random image → 0% fault detection rate
> - Different-round image (same activity) → 24% fault detection rate
>
> These findings support two conclusions:
> - **Image optimization is crucial**—it enhances the harmfulness of the input beyond a generic or random image.
> - **Image-text alignment matters**—even within the same activity, mismatched images and prompts reduce attack effectiveness.
> This validates our design choice to jointly optimize both modalities in our red-teaming framework.
>
> ---
> >**Q3: Missing comparisons with recent state-of-the-art methods.**
>
> R3: We appreciate the reviewer’s suggestion. In fact, our experiments are conducted on four open-source VLMs and one commercial model. The results consistently demonstrate that our TRUST-VLM framework achieves similarly strong performance across these diverse model architectures, supporting its generalizability.
>
> Additionally, we have compared our method with the state-of-the-art red-teaming method Arondight using the Qwen-VL model as the target. Due to space constraints, we kindly refer the reviewer to our response to Reviewer iB5W’s Q3 for details of this comparison. We will include these cross-model results and further clarify them in the revised version.
>
> ---
> >**Q4: Insufficient test cases and Lack of evaluation details.**
>
> R4: We agree with the importance of thorough experimentation. Due to the fact that the baseline method (RedTeaming VLM) provides only 200 jailbreak test cases, we maintained consistency by generating 200 test cases using our TRUST-VLM method for fair comparison.
>
> Moreover, our red-teaming evaluation spans four open-source VLMs, one commercial model and six distinct harmful categories. For each model, we generate 200 success test cases (1k test cases in total). The metrics reported in our paper represent the average performance across these categories and models. Detailed category-wise performance is also provided in the appendix for transparency. We agree that multiple evaluation runs per category per model can offer more reliable and stable results. Currently, we are in the process of generating additional adversarial test cases and conducting further experiments to compute standard deviations and variance measures, thus assessing stability comprehensively. We will share these extended results with you as soon as possible in our next response.

---

> > ### Comment · Reviewer_Bw4z · 2025-04-02
> >
> > Thank you for your thoughtful and detailed response. I appreciate the effort you have put into addressing my concerns. After reviewing your clarifications and additional results, I am satisfied that my concerns have been adequately addressed. The improvements and insights provided strengthen the paper, and I am happy to raise my score accordingly. Kindly include the promised results as well.

---

> > > ### Author Response · Authors · 2025-04-03
> > >
> > > Thank you for your detailed review and for acknowledging our rebuttal with an increased score. We greatly appreciate your insights and will reflect the discussed improvements in our paper revision.

---

### Official Review · Reviewer_iB5W · 2025-03-14

**Overall Recommendation:** 3

**Summary:**

This paper proposes a novel red-teaming framework (TRUST-VLM) to systematically uncover safety vulnerabilities in VLMs with black-box access. The proposed method improves both the fault detection rate and the diversity of generated test cases.
Extensive experiments show that TRUST-VLM not only outperforms traditional red-teaming techniques in generating diverse and effective adversarial cases but also provides actionable insights for model improvement.

**Claims And Evidence:**

Strengths:
- This paper proposes TRUST-VLM for the systematic identification of vulnerabilities in VLMs.
- The authors perform extensive experiments to demonstrate the effectiveness of the proposed TRUST-VLM.

Weaknesses:
- The paper doesn't adequately distinguish TRUST-VLM from existing adversarial attack methods. While the authors claim their approach is different from conventional adversarial attacks (in Table 1), they don't clearly establish how their red-teaming methodology conceptually differs from or improves upon established attack frameworks.
- The evaluation methodology has limitations. The authors primarily evaluate their method on fault detection rates and diversity metrics, but don't provide a thorough comparison with state-of-the-art jailbreaking techniques. Additionally, they exclude Arondight from their baseline comparisons due to a lack of open-source code, which weakens their comparative analysis.
- The paper provides limited discussion of defensive methods. While the focus is on identifying vulnerabilities, there's minimal discussion about how to mitigate the identified issues, reducing the practical impact of the research.

**Essential References Not Discussed:**

No.

**Experimental Designs Or Analyses:**

- The paper doesn't adequately distinguish TRUST-VLM from existing adversarial attack methods. While the authors claim their approach is different from conventional adversarial attacks (in Table 1), they don't clearly establish how their red-teaming methodology improves upon established attack frameworks.
- The evaluation methodology has limitations. The authors primarily evaluate their method on fault detection rates and diversity metrics, but don't provide a thorough comparison with state-of-the-art jailbreaking techniques.

**Methods And Evaluation Criteria:**

Yes.

**Other Comments Or Suggestions:**

No

**Other Strengths And Weaknesses:**

No

**Questions For Authors:**

No

**Relation To Broader Scientific Literature:**

The proposed method in this paper connects to the broader scientific literature on AI safety, multimodal systems, and adversarial testing.
- extending red-teaming methodologies from language-only models to vision-language models (VLMs).
- introducing a novel feedback mechanism that uses the target model's responses to iteratively improve attack strategies.

**Theoretical Claims:**

This paper does not involve the claim and proof of novel theories.

---

> ### Author Rebuttal · Authors · 2025-04-01
>
> Thanks for your careful review and thoughtful comments.
>
> >**Q1: Distinction from existing adversarial methods.**
>
> R1: We apologize for any confusion caused. Our red-teaming method differs significantly from adversarial attacks such as jailbreak. As detailed in Related Works (Section 2.3), the primary objective of red-teaming is to systematically explore and uncover a broader and more diverse set of vulnerabilities in VLMs. Red-teaming covers a wide range of inputs, including deliberate adversarial attacks (such as jailbreak), as well as unintentional harmful prompts from regular users—essentially any input that could potentially lead to harmful outputs. In contrast, adversarial attacks like jailbreak focus primarily on maximizing the attack success rate by generating specific, malicious prompts, inherently limiting their capacity to discover diverse model vulnerabilities.
>
> We believe that adversarial attacks such as jailbreak complement our red-teaming approach and can together help model developers better assess the safety of their models. As outlined in our response to the usage of red-teaming in safety defense, our TRUST-VLM method provides actionable adversarial test cases that directly support developers in refining and aligning their VLMs, further highlighting the practical value and distinct advantage of our red-teaming approach.
>
> >**Q2: Comparison with SOTA jailbreak attacks.**
>
> R2: Thank you for pointing this out. To further illustrate the distinction between our red-teaming framework and traditional jailbreak-style attacks, we briefly compare our method with two recent SOTA jailbreak approaches below:
> - Qi et al. [1] adapt adversarial attacks from the computer vision domain to VLMs by applying PGD-based perturbations to clean images. The attack input is structured as {image: random clean image + perturbation, text: random harmful prompt}, with the goal of inducing harmful responses. However, since there is no meaningful correlation between images and texts, and the attack only perturbs the image, the success rate remains very low, as shown in our experimental comparison below. Moreover, the image inputs are fixed and lack diversity.
> - Li et al. [2] improve the attack success rate by jointly optimizing both the image and text inputs. While this boosts effectiveness, the resulting test cases tend to be extremely toxic and can easily be filtered out by safety filters—undermining their utility in real-world red-teaming.
>
> | Methods | Average Fault Detection Rate | Average Toxicity          | Average Diversity         | Average Alignment         |
> |---------|------------------------------|----------------------------|----------------------------|----------------------------|
> | VAEJA   | 66%                          | /                          | /                          | /                          |
> | HADES   | 100%                         | text: 83%, visual: 99%     | text: 0.91, visual: 0.32   | text: 0.45, visual: 0.27   |
> | Ours    | 99%                          | text: 12%, visual: 51%     | text: 0.88, visual: 0.50   | text: 0.76, visual: 0.26   |
>
> In contrast, our TRUST-VLM framework generates realistic, diverse, and semantically aligned image-text test cases, making it more suitable for comprehensive safety evaluation and alignment.
>
> >**Q3: Limited baseline comparison due to Arondight exclusion.**
>
> R3: We fully agree that Arondight is highly relevant and would serve as a valuable baseline for comparison. In fact, we recognized its importance during the initial stages of our experiments and made efforts to include it. Unfortunately, since the authors have not released the official code and the prompts used in their paper, we are unable to reproduce their method.
> To provide a preliminary comparison in the meantime, we have directly compared the reported Arondight results under a shared experimental setting—using Qwen-VL as the target model and three overlapping harmful categories: Illegal Activity, Adult Content, and Violent Content. Moreover, the metrics used in both works are also the same: fault detection rate and the prompt diversity. As shown in the table, our method achieves superior performance in both fault detection rate and prompt diversity.
>
> | Methods    | Illegal Activity | Adult Content | Violent Content | Average Diversity |
> |------------|------------------|----------------|------------------|-------------------|
> | Arondight  | 82%              | 35%            | 92%              | 0.58              |
> | Ours       | 98%              | 98%            | 100%             | 0.88              |
>
>
>
> >**Q4: Limited discussion on defensive methods.**
>
> R4: Due to space constraints, we kindly refer you to our detailed responses to Reviewer YghG’s Q3.
>
> ---
> [1] Qi et al. Visual Adversarial Examples Jailbreak Aligned Large Language Models
>
> [2] Li et al. Images are Achilles' Heel of Alignment: Exploiting Visual Vulnerabilities for Jailbreaking Multimodal Large Language Models

---

### Decision · Program_Chairs · 2025-05-01

**Decision:**

Accept (poster)

**Comment:**

The paper proposes TRUST-VLM, an automated red-teaming framework for Vision-Language Models that leverages in-context learning and model feedback to enhance attack effectiveness and diversity. While reviewers initially raised concerns about novelty and evaluation breadth, the rebuttal provided thorough clarifications and additional results, leading all reviewers to raise or maintain their positive assessments. Overall, the paper is viewed as a valuable and timely contribution to VLM safety research.